# HRN: A Holistic Approach to One Class Learning

Wenpeng Hu[1,*], Mengyu Wang[2,*], Qi Qin[2,3] , Jinwen Ma[1], and Bing Liu[2,†]

[1]Department of Information Science, School of Mathematical Sciences, Peking University
[2]Wangxuan Institute of Computer Technology, Peking University
[3]Center for Data Science, AAIS, Peking University
*{wenpeng.hu,wangmengyu,qinqi,jwma,dcsliub}@pku.edu.cn*

## Abstract

Existing neural network based one-class learning methods mainly use various forms of auto-encoders or GAN style adversarial training to learn a latent representation of the given one class of data. This paper proposes an entirely different approach based on a novel regularization, called *holistic regularization* (or *H-regularization*), which enables the system to consider the data holistically, not to produce a model that biases towards some features. Combined with a proposed *2-norm instance-level data normalization*, we obtain an effective one-class learning method, called HRN. To our knowledge, the proposed regularization and the normalization method have not been reported before. Experimental evaluation using both benchmark image classification and traditional anomaly detection datasets show that HRN markedly outperforms the state-of-the-art existing deep/non-deep learning models. The code of HRN can be found here[3].

## 1   Introduction

One-class learning or classification has many applications. For example, in information retrieval, one has a set of documents of interest and wants to identify more such documents [55]. Perhaps, the biggest application is in anomaly or novelty detection, e.g., intrusion detection, fraud detection, medical anomaly detection, anomaly detection in social networks and Internet of things, etc [8, 9]. Recently, image and video based applications have also become popular [13, 49, 70]. More details about these applications and others can be found in the recent survey [7, 61].

**One-class learning**: Let $\mathcal{X}$ be the space of all possible data. Let $X \subseteq \mathcal{X}$ be the set of all instances of a particular class. Given a training dataset $T \subseteq X$ of the class, we want to learn a one-class classifier $f(x) : \mathcal{X} \rightarrow \{0, 1\}$, where $f(x) = 1$ if $x \in X$ (i.e., $x$ is an instance of the class) and $f(x) = 0$ otherwise (i.e., $x$ is not an instance of the class, e.g., an *anomaly*). In most applications, deciding whether a data instance belongs to the given training class or is an anomaly can be subjective and a threshold is often used based on the application. Like most existing papers [68, 64, 8, 82], this work is interested in a score function instead, and ignores the above binary decision problem. In this case, the commonly used evaluation metric is AUC (Area Under the ROC curve).

Early works on one-class classification or learning include one-class SVM (OCSVM) [75], and Support Vector Data Description (SVDD) [78]. More recently, deep learning models have been proposed for the same purpose [68, 8], which mainly learn a good latent representation of the given

class of data using various auto-encoders [1, 14, 65, 71, 83, 69] or GAN [27] style adversarial training [74, 72, 15, 64]. Recent surveys of one-class classification can be found in [8, 37].

In this paper, we propose an entirely new one-class learning approach, which directly learns from a single class of data without using any auto-encoder or adversarial training technique. The key novelty of the proposed method is a new loss function (called *one-class loss*), which consists of *negative log likelihood* (NLL) for one class and a novel regularization method called *holistic regularization* (or *H-regularization*). This new regularization constrains the model training so that it considers the one class of data holistically, not arbitrarily biases any features. We argue that one of the key issues of one-class learning is how to avoid biasing some features in model building as we have no idea where anomalies or negative data may be or what their distribution may be. Any bias can be detrimental. This issue has not been explicitly addressed by existing approaches. Combined with a *2-norm instance-level normalization* for each data instance (different from that in [79], see Sec. 3.2), we obtain an effective one-class learning method, called **HRN** (*H-Regularization with 2-Norm instance-level normalization*). To our knowledge, both H-regularization and the normalization method have not been reported in the literature. Empirical evaluation using three image classification datasets widely used in evaluating one-class learners and three traditional benchmark anomaly detection datasets demonstrates the effectiveness of HRN. It outperforms eleven state-of-the-art baselines considerably.

On broader impact, we believe that our holistic one-class learning can help *positive and unlabeled* (PU) *learning* [52], *open-world learning* (or *out-of-distribution detection*) [23], and *continual learning* [11, 63] as all these learning paradigms need to face unseen/novel situations. We will briefly discuss a continual learning method based on the proposed one-class loss, which achieves very good results.

## 2 Related Work

Much of the existing work on anomaly, outlier or novelty detection can be regarded as some form of one-class learning from a class of normal data. Early work in statistics [4] was mainly based on probabilistic modeling of the distribution of the normal data and regard data points with low probabilities in the distribution as anomalies [4, 87, 22, 86]. In general, anomaly detection algorithms can be classified into the following categories: distance based methods [42, 3, 28, 31, 60], density based methods [38, 56, 6], mixture models [3, 43], one-class classification based methods [75, 78, 39], deep learning based representation learning using auto-encoders [10, 89, 69, 7, 94] and adversarial learning [74, 15, 64, 20], ensemble methods [53, 10], graphs and random walks [58, 31], transfer learning [45, 2], and multi-task learning [35]. Several surveys have also been published [9, 66, 7, 61].

About one-class learning, one-class SVM (OCSVM) [75] was perhaps the earliest method, which uses the kernel SVM to separate the data from the origin. It essentially treats the origin as the only negative data point. Another earlier method based on kernel SVM is the *Support Vector Data Description* (SVDD) [78], which tries to find a hypersphere to enclose the given class of data. [21] learns features using deep learning and then applies OCSVM or SVDD to build the one-class model.

The recent DSVDD (Deep Support Vector Data Description) proposed a deep learning solution to implement SVDD [68]. Similar to the original SVDD, it trains a neural network to minimize the volume of the hypersphere that encloses the given class of data. Our HRN system does not use these ideas and it outperforms OCSVM and DSVDD significantly (see Sec. 4). Most deep learning anomaly/novelty detection methods are based on one-class learning. They almost exclusively use the neural network representation learning capability to generate a latent representation of the given class [1, 14, 16, 25, 51, 65, 68, 71, 83, 92, 95, 69, 73, 26, 82]. Most methods employ various forms of auto-encoders. Some also use GAN [27] based methods [72, 64, 93, 20]. Some even use anomalies in the training data to build multi-class classifiers [76, 36, 62]. Additionally, there are works based on neural density estimation [81], multiple hypothesis prediction [59], robust mean estimation [19], etc. For a survey of deep learning based one-class anomaly detection methods, see [8]. Our work is different as we do not use an auto-encoder, adversarial training, or any other above method.

OCGAN [64] is a representative work on one-class anomaly detection using both an auto-encoder and a GAN style adversarial learning. It first uses an auto-encoder to learn a latent representation of the given class. It then forces latent representations of in-class normal examples to be distributed uniformly across the latent space. Finally, it trains a discriminator using the GAN's adversarial learning to differentiate between images of the given class and fake images generated from random latent samples using its decoder. When the discriminator is fooled, fake images chosen at random in

general will look similar to examples from the given class. Then the latent representation generated for the given class is of good quality. Earlier GAN-based methods include [74, 72, 15].

Also related is the out-of-distribution discovery. The in-distribution data may consist of 5 classes of CIFAR10 and is used to build a model, which is tested using another class not used in training. Various forms of thresholding were used to detect anomalies [23, 24, 33, 51, 18, 76, 36, 20, 85].

# 3 Proposed HRN Model

**Background**: In general, a supervised machine learning model is trained to minimize the expected error over the training data, known as *empirical risk minimization*. That is, given the training data $\mathbf{X}$ and its corresponding label set $Y$, a model $f(\cdot)$, parameterized by $\theta$, is trained to minimize the error (or loss) between $f(\mathbf{X})$ and $Y$:

$$\min_{\theta} \mathcal{L}(f(\mathbf{X}), Y), \tag{1}$$

where $\mathcal{L}(\cdot)$ is the loss function. With the help of the loss function and an optimization method, model $f(\cdot)$ can be learned to map $\mathbf{X}$ to $Y$. An important requirement of this classic supervised learning paradigm is that it needs at least two classes of data in order to learn.

However, in our case, we only have a single class of data. Here we present the proposed one-class learning method HRN, which uses the above learning paradigm, but employs a novel loss function called *one-class loss* with an accompanied instance-level data normalization method.

The architecture of $f(\cdot)$ can be any existing neural network. This paper uses a simple multilayer perceptron (MLP) with a single output unit, which already achieves very good results. Formally, the $i^{th}$ layer of the MLP is:

$$\mathbf{y}_i = \sigma(\mathbf{x}_i) \tag{2}$$

where $\sigma$ is the activation function. We suggest to use ReLU or Leaky-ReLU (see Sec. 3.1).[4] $\mathbf{x}_i$ is the input of the current layer (output of the last layer) or is $\mathbf{x}$ if the current layer is the first layer. Note that no activation function is used in the final layer (a single output unit) as a Sigmoid function is applied on $f(\cdot)$ to squash the output to $(0, 1)$ during training.

## 3.1 One-class Loss

In learning the given class $C$ with its training data, the proposed one-class loss is:

$$\mathcal{L} = \underbrace{\mathbb{E}_{\mathbf{x} \sim \mathbb{P}_{\mathbf{x}}} \left[ -\log(\text{Sigmoid}(f(\mathbf{x}))) \right]}_{\text{NLL}} + \lambda \cdot \underbrace{\mathbb{E}_{\mathbf{x} \sim \mathbb{P}_{\mathbf{x}}} \|\nabla_{\mathbf{x}} f(\mathbf{x})\|_2^n}_{\text{H-regularization}} \tag{3}$$

where $\mathbb{P}_{\mathbf{x}}$ denotes the data distribution of class $C$, and exponent $n$ and $\lambda$ are hyper-parameters controlling the strength of the penalty and balancing the regularization respectively. $\text{Sigmoid}(f(\mathbf{x})) \in (0, 1)$ can be seen as the probability of $\mathbf{x}$ belonging to class $C$. Since we have only one class/head in the output, using $\text{Sigmoid}()$ is a natural choice. We explain the two terms in Eq. (3) below.

**NLL (*Negative Log Likelihood* for one-class)**. Minimizing NLL means to train the model $f(\cdot)$ to output high values (thus low NLL) for the input training data of the class according its distribution to help recognize instances belonging to the given class. However, since we only have one class of data, minimizing NLL leads to two major problems:

*Problem-I (uncontrollable $f(\mathbf{x})$ output)*. It may lead to a saturated $\text{Sigmoid}(f(\cdot))$ which means that $\text{Sigmoid}(f(\cdot))$ will output 1 all the time. We have no control over the growth or the value of $f(\cdot)$ as Sigmoid flattens out after a certain value of $f(\cdot)$. Thus minimizing NLL (i.e., maximizing $f(\cdot)$) can lead to malformed parameters, e.g., all parameters may have large absolute values of arbitrary magnitudes, which results in the high chance that an anomaly or noise may get a very high $f(\cdot)$ value.

*Problem-II (feature bias)*. Features (or dimensions) of the input data with high values are very likely to be emphasized by the head and their related parameters are likely to have very high values. But those features might not be the important features for recognizing whether an input test instance belongs to the given class or not, which leads to poor accuracy. This problem is caused by the fact that we don't have other classes to compare with to identify the most discriminative features.

**H-regularization (holistic regularization).**[5] H-regularization aims to solve these two problems. For Problem-I, assume the head for class $C$ is a two-layer MLP with a single output unit (which is the case in HRN) and $\sigma(\cdot)$ is the activation function. Then, we can show $f(\mathbf{x}) = \mathbf{w}_2 \cdot \sigma(\mathbf{w}_1 \mathbf{x})$, where $\mathbf{w}_1$ and $\mathbf{w}_2$ are the parameters of the first and second layer respectively. Thus, we have:

$$\mathop{\mathbb{E}}_{\mathbf{x} \sim \mathbb{P}_\mathbf{x}^C} \|\nabla_\mathbf{x} f(\mathbf{x})\|_2^n = \mathop{\mathbb{E}}_{\mathbf{x} \sim \mathbb{P}_\mathbf{x}^C} \|\mathbf{w}_2 \cdot \nabla_{\mathbf{w}_1 \mathbf{x}} \sigma(\mathbf{w}_1 \mathbf{x}) \cdot \mathbf{w}_1\|_2^n. \tag{4}$$

The exact expression depends on the activation function. For ReLU (which we use in HRN), the elements in $\nabla_{\mathbf{w}_1 \mathbf{x}} \sigma(\mathbf{w}_1 \mathbf{x})$ are either 1 (ReLU$(\mathbf{w}_1 \mathbf{x}) > 0$) or 0 (ReLU$(\mathbf{w}_1 \mathbf{x}) \leq 0$). Let us first consider $\nabla_{\mathbf{w}_1 \mathbf{x}} \sigma(\mathbf{w}_1 \mathbf{x}) \equiv 1$ for all elements, which gives us:

$$\mathop{\mathbb{E}}_{\mathbf{x} \sim \mathbb{P}_\mathbf{x}^C} \|\nabla_\mathbf{x} f(\mathbf{x})\|_2^n = \|\mathbf{w}_2 \cdot \mathbf{w}_1\|_2^n. \tag{5}$$

Clearly, H-regularization can constrain the arbitrary growth of $\mathbf{w}_1$ and $\mathbf{w}_2$ parameter values and consequently the arbitrary growth and magnitude of $f(\cdot)$ because the arbitrary growth of the parameter values will lead to high penalties on H-regularization and thus high losses, i.e., a trade-off between NLL and H-regularization. Specifically, a high parameter value leads to a high $f(\cdot)$ and thus a low NLL, but a high value for H-regularization. The training goal of the one-class loss is thus to find a point where $f(\cdot)$ outputs a value as high as possible under the condition of having parameters with values as small as possible. Equivalently, it is to achieve Sigmoid$(f(\cdot))$ close to 1 while $f(\cdot)$ as small as possible. This is achievable as Sigmoid$(f(\cdot))$ flattens out after $f(\cdot)$ reaches a certain value.

When $\nabla_{\mathbf{w}_1 \mathbf{x}} \sigma(\mathbf{w}_1 \mathbf{x}) \equiv 1$ for all elements is not true, the 0 elements in it simply block some neurons/units, which we can ignore because the blocked neurons have no contributions to the final $f(\cdot)$ output. Note that we suggest to use piece-wise linear function as the activation function, e.g., ReLU and Leaky-ReLU, as both Sigmoid and Tanh are too flat for high input values. Take Sigmoid as an example, $\nabla_{\mathbf{w}_1 \mathbf{x}} \sigma(\mathbf{w}_1 \mathbf{x}) = \sigma(\mathbf{w}_1 \mathbf{x})(1 - \sigma(\mathbf{w}_1 \mathbf{x}))$, if $\mathbf{w}_1$ is already biased (with high values), the regularization tends to be blocked.

For Problem-II, as we know, the derivative $\nabla_\mathbf{x} f(\mathbf{x})$ shows the importance of each feature of $\mathbf{x}$. The features with large derivatives contribute more to the final output as small changes in them can lead to large changes in the $f(\mathbf{x})$ output and they also give large values for H-regularization, which is undesirable for loss minimization. In this case, minimizing H-regularization can ease the problem that the output is dominated by some specific features of the input $\mathbf{x}$.

We can also reach this conclusion using Eq. (5), the dimensions in $\mathbf{w}_2 \cdot \mathbf{w}_1$ corresponding to the contributions of the same feature dimensions of the input. In this case, the output will not be saturated by a few features of the input due to the H-regularization expressed as the right-hand-side of Eq. (5). In addition to this, since the L2-norm in Eq. (5) gives more penalties to the features with high values and little penalty to the features with low values, the parameter values will be more balanced. Note, we give the proposed regularization its name because it constrains the model to consider the input data more holistically rather than being biased by some specific features and noises in the data.

Note that the Gradient Penalty (GP) in WGAN [29] is defined as $\mathbb{E}_{\hat{\mathbf{x}} \sim \mathbb{P}_{\hat{\mathbf{x}}}}[(\|\nabla_{\hat{\mathbf{x}}} f(\hat{\mathbf{x}})\|_2 - 1)^2]$ to make $f(\cdot)$ a 1-Lipschtiz function, which looks similar to our H-regularization. However, it behaves differently especially when $\|\nabla_{\hat{\mathbf{x}}} f(\hat{\mathbf{x}})\|_2 < 1$ (which has an opposite effect to ours), and is thus not suitable for our work. We experimented with it and got poor results.

### 3.2 2-Norm Instance-Level Data Normalization

Different feature scales in data instances can lead to different output scales of $f(\cdot)$, which may confuse the model to produce poor results. Let an input data instance be $\mathbf{x}$ and its 2-norm be $\|\mathbf{x}\|_2$. Assume the model $f(\cdot)$ is a two-layer MLP with a single output unit and ReLU is the activation function (as suggested in Sec. 3.1 and used in HRN). It is easy to see $f(\mathbf{x}) = \mathbf{w}_2 \cdot \text{ReLU}(\mathbf{w}_1 \mathbf{x})$, where $\mathbf{w}_1$ and $\mathbf{w}_2$ are the parameters of the first and second layer respectively, and

$$\|f(\mathbf{x})\|_2 = \|\mathbf{w}_2 \cdot \text{ReLU}(\mathbf{w}_1 \mathbf{x})\|_2 \leq \|\mathbf{w}_2\|_2 \cdot \|\text{ReLU}(\mathbf{w}_1 \mathbf{x})\|_2$$
$$\leq \|\mathbf{w}_2\|_2 \cdot \|\mathbf{w}_1 \mathbf{x}\|_2 \leq \|\mathbf{w}_2\|_2 \cdot \|\mathbf{w}_1\|_2 \cdot \|\mathbf{x}\|_2. \tag{6}$$

This derivation uses consistent matrix norm properties $\|AB\|_2 \leq \|A\|_2 \|B\|_2$ and $\|\text{ReLU}(\mathbf{x})\|_2 \leq \|\mathbf{x}\|_2$. Eq. (6) shows the scale of $\mathbf{x}$ can affect the upper bound of $f(\mathbf{x})$. Given $\mathbf{x}$ with a large norm, we tend to get a high $f(\cdot)$ response.

To deal with this issue, we normalize $\mathbf{x}$ so that its norm is 1, i.e., $\mathbf{x} := \mathbf{x}/\|\mathbf{x}\|_2$, which we call *2-norm instance normalization*. This is an instance-level normalization, which is different from the traditional feature-level normalization that normalizes each feature across all instances.

We further subtract the mean from each feature value to make the feature values of each instance having zero-mean. Without this subtraction, all positive feature values in the input data will result in all parameters of $f(\cdot)$ positive (see Eq. 3). With negative values in the input data, some network parameter values can be negative, which increase the value space of parameters and consequently the probability of learning a better model. Note that this normalization is different from the instance normalization in [79], which is similar to the traditional z-score and normalizes the contrast of the images. It performs significantly poorer than our normalization (see Sec. 4.4).

## 4 Empirical Evaluation

We empirically evaluate the proposed algorithm HRN using six benchmark datasets and eleven state-of-the-art baselines. Following existing papers, no pre-trained feature extractors were used in the main evaluation. At the end of Sec. 4.3, we will try an ImageNet pre-trained feature extractor to see whether pre-training makes a difference. It can make a big difference. As a broader impact, Sec. 4.5 briefly describes a continual learning method that applies the proposed one-class loss.

### 4.1 Experiment Datasets and Baselines

**Datasets**. We use three benchmark image classification datasets and three benchmark traditional non-image anomaly detection datasets that have been used in many previous papers. (1) **MNIST** [47][6] is a handwritten digit classification dataset of 10 digits, i.e., 10 classes. The dataset has 70,000 examples/instances, with the splitting of 60,000 for training and 10,000 for testing. (2) **fMNIST** (fashion-MNIST) [84][7] consists of a training set of 60,000 examples and a test set of 10,000 examples of 10 classes. Each example is a 28x28 grayscale fashion picture. (3) **CIFAR-10** [44][8] is also an image classification dataset consisting of 60,000 32x32 color images of 10 classes with the splitting of 50,000 for training and 10,000 for testing.

For each of these three image datasets, we use the training data of each class $C$ in the dataset in turn as the one class data to build a model and then test the model using the full test set of all classes. The rest of the classes except $C$ are anomalies. The three non-image datasets are:

(4) **KDDCUP99** [9] consists of 450000 training instances and 44021 test instances of two classes. The majority class (80% of the data) is regarded as the one class used in learning. (5) **Thyroid** [10] uses the version in TQM [81] with 3772 instances, 1839 for training and 1933 for testing. The hyperfunction class is treated as the novel class and the rest as the one class for learning. (6) **Arrhythmia** [11] uses the data split of normal and abnormal in DAGMM [95] with 193 casee for training and 259 for testing.

**Baselines.** The following 11 baselines are compared. (1) **DSVDD** (Deep SVDD) [68]: A recent deep one-class classifier described in Sec. 2. (2) **ICS** [73]: A latest one-class method. It first splits the one class training data into two subsets, typical and atypical. It then trains a binary classifier. (3) **OCGAN** [64]: A latest GAN-based one-class anomaly detection method described in Sec. 2. (4) **ADGAN** [15]: Also a GAN-based method. GAN runs on the normal data to learn a generator to produce a latent representation to approximate the one class distribution. In testing, if a test case is drawn from the given one class, there exist some points in GAN's latent space which, after passing through the generator network, should generate something closely resemble this instance. ADGAN uses this idea to perform anomaly detection. (5) **TQM** [81]: A latest work that produces a multivariate *triangular quantile maps* (TQM) score function, which is estimated using an auto-encoder, a flow-based neural density estimator, and KL-divergence. (6) **DAGMM** [95]: A joint model using an auto-encoder and a Gaussian mixture model. It first uses the auto-encoder to generate a latent

representation and the reconstruction error for each input data instance, which are then fed into the Gaussian Mixture Model. (7) **VAE** [40]: The classic variational auto-encoder. As mentioned in Sec. 2 that many existing works are based auto-encoders, we thus include this and the next auto-encoder baselines. (8) **DAE** (Denoising auto-encoder) [30, 80]: The reconstruction error is used as the scoring function. (9) **OCSVM** [75]: A well-known one-class SVM method based on kernel SVM (see Sec. 2) (10) **iForest** [53]: An ensemble method that builds a number of random unsupervised trees to isolate anomalies. (11) **HRN-L2**: HRN with H-regularization replaced by L2-regularization.

## 4.2 Training Details and Hyper-parameter Selection

HRN uses a simple MLP (Multilayer Perceptron), which can produce the state-of-the-art results. Each experiment on a class takes less than 5 minutes. Specifically, a MLP of size [784-100]-[100-1] is used for MNIST and fMNIST, of size 3*[1024-300]-[900-300]-[300-1] for CIFAR-10 [12] and of size [125-100]-[100-1] for KDDCUP99, and of size [6-100]-[100-1] for Thyroid. We use SGD with moment as the optimizer. The learning rate is 0.1. To get the best of baselines, we take their results from their papers whenever possible as the experiment setups are the same and only run their code when a result was not reported. In each experiment, we run HRN 100 epochs and run baselines using their original settings to get the maximum accuracy. We repeat this 5 times and report the average result. Hyper-parameter tuning for one-class learning is more challenging as there is only one class of data. TQM [82] set 10% of the data as the validation set for each dataset. DSVDD [68] reported the best test result using a range of values for its hyper-parameters. Two hyper-parameters in HRN need tuning, $\lambda$ and $n$ in H-regularization (Eq. 3). We followed the TQM approach and used grid search. However, we used only the MNIST data to search for hyper-parameter values and then applied the values to all 5 datasets. Grid search uses the following tuning ranges: for $\lambda$, from 0 to 1 with step 0.05 and for $n$, from 1 to 20 with step 1. This gives 41 combinations as we try one, fix it and then the next. After tuning on MNIST, we get $\lambda = 0.1$ and $n = 12$ (the results are similar for $10 \leq n \leq 16$), which were applied to all datasets in all experiments without change.

## 4.3 Results and Discussion

**Image Datasets:** We first report the results on the three image datasets and then the three non-image datasets. The main results on the three image datasets are given in Table 1.[13] AUC (*Area Under the ROC curve*) is the evaluation metric, which is also used in most one-class and other anomaly detection algorithms [7, 64, 68]. Each row in the table gives the average results over 5 runs of all compared systems using one class (column 1) as the training data for a dataset. The last row gives the average result of each column.

Table 1 allows us to make the following observations. **(1)** on average, the proposed algorithm HRN outperforms all baselines consistently. **(2)** TQM does quite well for MNIST and fMINIST (although it is slightly worse than HRN). However, it performs poorly on CIFAR-10. On average, its AUC is only 52.10 while our HRN's AUC is 71.32. This dataset was not used in the experimental evaluation of the TQM paper. We used the code released by the authors and were able to reproduce their results on MNIST and fMNIST. We tried to modify and optimize it for CIFAR-10, but were not able to obtain a better result. **(3)** Overall ICS is the strongest baseline, but is still significantly weaker than HRN. Its average AUC on the CIFAR-10 dataset is much better than other baselines, although it is still markedly lower than HRN. **(4)** DAGMM is very weak for these datasets as it was not designed for image data. **(5)** OCGAN is very competitive on MNIST, but does not do well on the other two datasets compared to HRN. OCSVM and iForest also did reasonably well, but pooer than those deep learning based methods in general. **(6)** MNIST is the easiest dataset and most systems do fairly well except DAGMM. CIFAR-10 is the hardest and HRN does considerably better than the baselines. **(7)** HRN-L2 (HRN with L2-regularization) is markedly poorer than HRN. One reason is that there is still a very high $\mathbb{E}_{\mathbf{x} \sim \mathbb{P}_{\mathbf{x}}} \|\nabla_{\mathbf{x}} f(\mathbf{x})\|_2$, e.g., up to 9.37 on MNIST, but only 0.872 when optimizing H-regularization, which shows that the output of $f(\cdot)$ is sensitive to the input $\mathbf{x}$. For example, $\Delta \mathbf{x} = 0.1$ will lead to an output change of up to 0.937 which is much higher than 0.087 for H-regularization.

Table 1: Average AUCs in % over 5 runs per method on the three image datasets.

| Class | OCSVM | iForest | DAE | VAE | DAGMM | ADGAN | OCGAN | DSVDD | ICS | TQM | HRN-L2 | HRN |
|---|---|---|---|---|---|---|---|---|---|---|---|---|
| **MNIST** | | | | | | | | | | | | |
| 0 | 98.6 | 96.9 | 89.4 | 99.7 | 50.0 | 99.5 | **99.8** | 98.0 | 98.9 | 99.5 | 97.0 | 99.5±0.0 |
| 1 | 99.5 | 99.5 | 99.9 | 99.9 | 76.6 | 99.9 | 99.9 | 99.7 | 99.8 | 99.8 | 98.7 | **99.9**±0.0 |
| 2 | 82.5 | 75.6 | 79.2 | 93.6 | 32.6 | 93.6 | 94.2 | 91.7 | 91.7 | 95.3 | 89.4 | **96.5**±0.1 |
| 3 | 88.1 | 83.5 | 85.1 | 95.9 | 31.9 | 92.1 | 96.3 | 91.9 | 96.6 | 96.3 | 92.3 | **97.4**±0.1 |
| 4 | 94.9 | 87.9 | 88.8 | 97.3 | 36.8 | 94.9 | **97.5** | 94.9 | 86.5 | 96.6 | 91.6 | 97.2±0.1 |
| 5 | 77.1 | 75.5 | 81.9 | 96.4 | 49.0 | 93.6 | **98.0** | 88.5 | 88.9 | 96.2 | 76.2 | 97.2±0.2 |
| 6 | 96.5 | 87.4 | 94.4 | 99.3 | 51.5 | 96.7 | 99.1 | 98.3 | 98.8 | 99.2 | 94.4 | **99.2**±0.0 |
| 7 | 93.7 | 90.6 | 92.2 | 97.6 | 50.0 | 96.8 | **98.1** | 94.6 | 96.1 | 96.9 | 92.0 | 97.6±0.1 |
| 8 | 88.9 | 73.8 | 74.0 | 92.3 | 46.7 | 85.4 | 93.9 | 93.9 | 95.0 | **95.5** | 90.7 | 94.3±0.2 |
| 9 | 93.1 | 88.0 | 91.7 | 97.6 | 81.3 | 95.7 | **98.1** | 96.5 | 90.0 | 97.7 | 91.4 | 97.1±0.0 |
| Avg | 91.29 | 85.87 | 87.66 | 96.96 | 50.64 | 94.82 | 97.50 | 94.80 | 94.23 | 97.30 | 91.37 | **97.59** |
| **fMNIST** | | | | | | | | | | | | |
| 0 | 86.1 | 91.0 | 86.7 | 87.4 | 42.1 | 89.9 | 85.5 | 79.1 | 88.3 | 92.2 | 91.5 | **92.7**±0.0 |
| 1 | 93.9 | 97.8 | 97.8 | 97.7 | 55.1 | 81.9 | 93.4 | 94.0 | **98.9** | 95.8 | 97.6 | 98.5±0.1 |
| 2 | 85.6 | 87.2 | 80.8 | 81.6 | 50.4 | 87.6 | 85.0 | 83.0 | 88.2 | **89.9** | 88.2 | 88.5±0.1 |
| 3 | 85.9 | 93.2 | 91.4 | 91.2 | 57.0 | 91.2 | 88.1 | 82.9 | 92.1 | 93.0 | 92.7 | **93.1**±0.1 |
| 4 | 84.6 | 90.5 | 86.5 | 87.2 | 26.9 | 86.5 | 85.8 | 87.0 | 90.2 | **92.2** | 91.0 | 92.1±0.1 |
| 5 | 81.3 | 93.0 | 92.1 | 91.6 | 70.5 | 89.6 | 88.5 | 80.3 | 89.4 | 89.4 | 71.9 | **91.3**±0.4 |
| 6 | 78.6 | 80.2 | 73.8 | 73.8 | 48.3 | 74.3 | 77.5 | 74.9 | 78.3 | **84.4** | 79.4 | 79.8±0.1 |
| 7 | 97.6 | 98.2 | 97.7 | 97.6 | 83.5 | 97.2 | 93.9 | 94.2 | 98.3 | 98.0 | 98.9 | **99.0**±0.0 |
| 8 | 79.5 | 88.7 | 78.2 | 79.5 | 49.9 | 89.0 | 82.7 | 79.1 | 88.6 | 94.5 | 90.8 | **94.6**±0.1 |
| 9 | 97.8 | 95.4 | 96.3 | 96.5 | 34.0 | 97.1 | 97.8 | 93.2 | 98.5 | 98.3 | **98.9** | 98.8±0.0 |
| Avg | 87.09 | 91.52 | 88.13 | 88.41 | 51.8 | 88.43 | 87.82 | 84.77 | 91.08 | 92.77 | 90.09 | **92.84** |
| **CIFAR-10** | | | | | | | | | | | | |
| 0 | 61.6 | 66.1 | 41.1 | 70.0 | 41.4 | 63.2 | 75.7 | 61.7 | 76.8 | 40.7 | 80.6 | **77.3**±0.2 |
| 1 | 63.8 | 43.7 | 47.8 | 38.6 | 57.1 | 52.9 | 53.1 | 65.9 | **71.3** | 53.1 | 48.2 | 69.9±1.3 |
| 2 | 50.0 | 64.3 | 61.6 | **67.9** | 53.8 | 58.0 | 64.0 | 50.8 | 63.0 | 41.7 | 64.9 | 60.6±0.3 |
| 3 | 55.9 | 50.5 | 56.2 | 53.5 | 51.2 | 60.6 | 62.0 | 59.1 | 60.1 | 58.2 | 57.4 | **64.4**±1.3 |
| 4 | 66.0 | 74.3 | 72.8 | 74.8 | 52.2 | 60.7 | 72.3 | 60.9 | **74.9** | 39.2 | 73.3 | 71.5±1.0 |
| 5 | 62.4 | 52.3 | 51.3 | 52.3 | 49.3 | 65.9 | 62.0 | 65.7 | 66.0 | 62.6 | 61.0 | **67.4**±0.5 |
| 6 | 74.7 | 70.7 | 68.8 | 68.7 | 64.9 | 61.1 | 72.3 | 67.7 | 71.6 | 55.1 | 74.1 | **77.4**±0.2 |
| 7 | 62.6 | 53.0 | 49.7 | 49.3 | 55.3 | 63.0 | 57.5 | **67.3** | 64.1 | 63.1 | 55.5 | 64.9±1.1 |
| 8 | 74.9 | 69.1 | 48.7 | 69.6 | 51.9 | 74.4 | 82.0 | 75.9 | 78.9 | 48.6 | 79.9 | **82.5**±0.2 |
| 9 | 75.9 | 53.2 | 37.8 | 38.6 | 54.2 | 64.2 | 55.4 | 73.1 | 66.0 | 58.7 | 71.6 | **77.3**±0.9 |
| Avg | 64.78 | 59.72 | 53.58 | 58.33 | 53.13 | 62.42 | 65.66 | 64.81 | 69.27 | 52.10 | 66.65 | **71.32** |

Table 2: Average precision, recall, and F1 score on the three non-image datasets over 5 runs

| | KDDCUP99 | | | Thyroid | | | Arrhythmia | | |
|---|---|---|---|---|---|---|---|---|---|
| Method | Precision | Recall | F1 | Precision | Recall | F1 | Precision | Recall | F1 |
| OCSVM | 74.57 | 85.23 | 79.54 | 36.39 | 42.39 | 38.87 | 53.97 | 40.82 | 45.81 |
| iForest | 1.0 | 89.88 | 94.67 | 99.88 | 89.57 | 94.44 | 83.90 | 89.12 | **86.43** |
| DAGMM | 92.97 | 94.42 | 93.69 | 47.66 | 48.34 | 47.82 | 49.09 | 50.78 | 49.83 |
| TQM | 96.22 | 96.22 | 96.22 | 75.27 | 75.27 | 75.27 | 53.03 | 53.03 | 53.03 |
| HRN | 98.83 | 98.83 | **98.83** | 95.87 | 95.87 | **95.87** | 84.46 | 84.46 | 84.46 |

**Non-Image Datasets:** Following the latest baseline TQM [81], we also use precision, recall and F1 score as the evaluation measures, and apply the same TQM's thresholding method in HRN, making the precision, recall and F1 scores the same. We use four baselines: OCSVM, iForest, DAGMM, and TQM, which can work on both image and non-image data (the other baselines were designed for images). The results are given in Table 2. HRN outperforms three baselines considerably. On the Thyroid dataset, the improvement is dramatic, from F1 of 75.27 (TQM) to 95.87 (HRN). HRN

Table 3: Average AUCs in % of different components of HRN on the image datasets. Hreg: H-regularization; 2N_Inst_Norm: our 2-norm instance normalization; Inst_Norm[79]: instance normalization in [79]; SquareLoss: Square Loss

|  | MNIST | fMNIST | CIFAR-10 |
|---|---|---|---|
| NLL | 55.42 | 57.12 | 52.64 |
| NLL+Hreg | 80.92 | 76.13 | 55.26 |
| NLL+Hreg+Inst_Norm[79] | 92.58 | 90.56 | 65.21 |
| NLL+Hreg+2N_Inst_Norm (HRN) | **97.59** | **92.84** | **71.32** |
| SquareLoss | 76.67 | 72.72 | 52.68 |
| SquareLoss+Hreg+2N_Inst_Norm (HRN) | 97.12 | 91.72 | 71.08 |

outperforms iForest on two datasets in F1, but is slightly poorer than iForest on Arrhythmia. This is because the Arrhythmia dataset is too small for deep learning, only 193 training instances.

**Using a Pre-trained Feature Extractor.** Pre-trained feature extractors have been shown to improve end-task results in many image applications [77]. In this experiment, we use CIFAR-10 as the evaluation dataset. We pre-train a WRN model [90] using ImageNet after manually removing 229 classes that are similar to those classes in CIFAR-10. We experimented pre-training with HRN and 3 top performing baselines. The average result over 10 classes for each system is reported in Table 4. Pre-training helps improve HRN drastically from 71.27 (Table 1) to 96.6 (Table 4). More details can be found in Supplementary Materials.

Table 4: Average AUCs in % on CIFAR-10: Pre-training using ImageNet without overlapping classes

| Method | OCGAN | ICS | TQM | HRN |
|---|---|---|---|---|
| AUC | 64.8 | 86.6 | 53.5 | **96.6** |

In summary, we believe that the superior performance of HRN is due to the fact that we argued that a key to good one-class learning is to avoid biasing any features in learning since we have no prior information where anomalies or negative data may be and we explicitly addressed this bias problem using H-regularization. Existing approaches did not explicitly deal with this problem.

## 4.4 Ablation Study and Additional Experiments

We now study the contributions of the two components of the HRN system. Here we also include the instance normalization method in [79]. Table 3 gives the ablation results of MNIST, fMNIST and CIFAR-10. We see that using only NLL, the model performed very poorly on all three datasets. Including H-regularization (NLL+Hreg) improves the performance drastically. When our 2-norm instance normalization was added (NLL+Hreg+2N_Inst_Norm), the results were improved even further. The instance normalization method in [79] (NLL+Hreg+Inst_Norm[79]) also did fairly well, but it is significantly poorer than HRN (NLL+Hreg+2N_Inst_Norm). This is because that the method in [79] does not solve the problem identified in Sec. 3.2.

**Square loss** can also be applied in place of NLL in Eq. 3. The last two rows of Table 3 show that square loss is better than NLL alone, but after adding Hreg+2N_Inst_Norm, NLL is slightly better.

**Training with Noise.** Here we study the robustness of the HRN model by assessing its performance under noisy training data. In learning each class, we randomly sample some examples from the other classes and add them to the training data of the class. We experimented with noise ratios: 1%, 10%, 20%. Table 5 gives the average AUC over all classes of each image dataset with different training noise ratios. We see that HRN can maintain very good performances with high levels of noise.

**Adversarial Attack.** To study the robustness of HRN against adversarial attack, we follow the DSVDD paper [68] using the 'stop sign' class of the German Traffic Sign Recognition Benchmark (GTSRB) dataset, and also its setup and method of generating adversarial examples. Table 6 shows the results of several strong models in Table 1. The results of GAN-based models are absent as [68] observed that GANs did not converge due to the small dataset size. HRN again outperforms the best performing baselines. Details can be found in Supplementary Materials.

Table 5: Average AUCs in % with different training data noise ratio for HRN.

| Noise ratio | MNIST | fMNIST | CIFAR-10 |
|---|---|---|---|
| 0% | 97.59 | 92.84 | 71.32 |
| 1% | 97.24 | 92.26 | 71.02 |
| 10% | 95.01 | 91.72 | 70.78 |
| 20% | 92.70 | 90.80 | 70.36 |

Table 6: Average AUCs in % per method on GTSRB stop sign adversarial attacks

| Method | DSVDD | ICS | TQM | HRN |
|---|---|---|---|---|
| AUC | 80.3 | 84.6 | 87.6 | **95.4** |

Table 7: Continual learning accuracy results for 1 class per task of HCL and the baselines.

| Dataset | w/o PFE | EWC | LwF | IMM | PGMA | RPSnet | OWM | HCL |
|---|---|---|---|---|---|---|---|---|
| MNIST (10 classes) | no | 9.91 | 19.96 | 29.16 | 71.36 | 40.29 | 94.46 | **97.00** |
| EMNIST-47 (47 classes) | no | 2.13 | 4.59 | 18.69 | 10.13 | 10.08 | 77.45 | **80.05** |
| DBPedia (14 classes) | no | 7.14 | 7.14 | 7.14 | 9.58 | 36.70 | 92.23 | **93.51** |
| DBPedia (14 tasks) | yes | 7.14 | 7.14 | 7.14 | 66.40 | 50.58 | 95.37 | **96.23** |

## 4.5 Continual Learning based on One-Class Classification

As mentioned in introduction, the proposed one-class learning may be applied to other learning paradigms. Here, we use it for *continual learning* (CL) of a sequence of tasks incrementally, where each task consists of one or more classes. CL mainly deals with *catastrophic forgetting* [57].

Our CL network (called HCL) uses a simple architecture. It consists of an optional pre-trained Feature Extractor (PFE) (not updated in learning each new task) shared by all tasks or classes, and Class Heads following it, one head for each class learned so far. Let the head for each class $C_i$ be $f_{C_i}(\cdot)$. Each head is an independent one-class model using a 2-layer MLP and a single output unit. When each new class (or task) comes, it is incrementally learned using the one-class loss (Sec. 3.1). The number of parameters for each one-class model is very small, which ensures that adding new tasks will not lead to a huge model. In testing, given a test instance $\mathbf{x}$, we choose the head that gets the highest $f_{C_i}(\cdot)$ output value as the class of $\mathbf{x}$, i.e.,

$$y = \underset{C_i}{\mathrm{argmax}}[f_{C_1}(\mathbf{x}_f), \ldots, f_{C_N}(\mathbf{x}_f)]. \tag{7}$$

where $N$ is the total number of classes learned so far; $\mathbf{x}_f$ is the feature obtained by the pre-trained feature extractor: $\mathbf{x}_f = \mathcal{F}(\mathbf{x})$, or the input data $\mathbf{x}$ itself when PFE is not used. We have conducted some initial experiments with one class per task continual learning.

**Datasets:** Two benchmark image classification datasets, **MNIST** [47] and **EMNIST-47** [12], and one text classification dataset **DBPedia** are used in our experiments.

**Baselines**: We compare with the following classic and the latest state-of-the-art class continual learning (CCL) baselines that do not use saved training examples from old tasks: (1) **EWC** [41], (2) **LwF** [50], (3) **IMM** [48], (4) **PGMA** [34], (5) **RPSnet** [67], and (6) **OWM** [91].

**Results**: Table 7 shows the accuracy results. HCL outperforms the baselines markedly with/without (w/o) PFE. DBPedia uses BERT [17]) as its PFE. MNIST and EMNIST-47 do not need PFE.

## 5 Conclusion

Existing approaches to one-class learning using deep learning are mainly based on GAN and auto-encoders to learn a latent representation of the given class. This paper proposed an entirely different approach called HRN, which uses a new *one-class loss* function with a novel regularization method. Combined with a 2-norm instance-level data normalization, we obtained a highly effective one-class learning model. The architecture of HRN is also very simple. Experimental results showed its superior performance compared to strong baselines. In our future work, we plan to investigate more complex architectures to further improve the accuracy of the proposed HRN method, and also exploit the one-class method for PU learning and open-world learning.

## Acknowledgments and Disclosure of Funding

This work was partially supported by the National Key Research and Development Program of China under grant 2018AAA0100205.

## Broader Impact

One-class learning has a wide range of applications, especially in anomaly or novelty detection, e.g., detecting intrusions, fraud, medical anomalies, and anomalies in social networks, Internet of things, text documents, images, and videos. Perhaps, more importantly, we believe that our holistic one-class learning can help positive and unlabeled (PU) learning, open-world learning (or out-of-distribution detection), and continual learning as all these learning paradigms need to face unseen/novel situations. We have shown a continual learning application in the paper. We don't see that anyone could be put at disadvantage from this research. The consequence of failure of the system is that the system recognizes some anomalies wrongly. We don't think that the task or the method leverages biases in the data. In fact, our proposed holistic regularization tries to avoid using any biases in the data.

## Footnotes

†Corresponding author. The work was done when B. Liu was at Peking University on leave of absence from University of Illinois at Chicago, liub@uic.edu.

[3]https://github.com/morning-dews/HRN

[4]Using a ReLu-like activation by no means a restriction as it is widely used, e.g., in Transformer, ResNet, etc.

[5] H-regularization has some resemblance to L2 regularization. We will see L2 is significant poorer in Sec. 4.3.

[6] http://yann.lecun.com/exdb/mnist/

[7] https://github.com/zalandoresearch/fashion-mnist

[8] https://www.cs.toronto.edu/ kriz/cifar.html

[9] http://kdd.ics.uci.edu/databases/kddcup99

[10] http://archive.ics.uci.edu/ml

[11] http://archive.ics.uci.edu/ml

[12]The first layer has 3 sub-modules for extracting features independently from the 3 channels of the CIFAR-10 images. We then concatenate the outputs of the three sub-modules as the input to the second layer.

[13]We also experimented with CIFAR100 (100 classes), which is not used in baseline papers. HRN gets the average AUC of 68.61 and the top baselines OCGAN and ICS get only 52.26 and 60.79 respectively.

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
