[Supplementary Material · Appendix.pdf]

# 6 Supplementary Materials

## 6.1 Adversarial Attack Experiments

Here we study the robustness of our model against adversarial attack. Adversarial attack is a technique that generates data instances with the intention to fool the model. In many applications (e.g., autonomous driving), it is vital to detect adversarial instances to ensure safety. We now examine how our HRN fares on this task.

**Experiment setup**. We follow the DSVDD paper [68] using the 'stop sign' class of the German Traffic Sign Recognition Benchmark (GTSRB) dataset and their setup in our experiment. Adversarial examples are generated from randomly drawn stop-sign images of the test set using Boundary Attack [5]. The training dataset consists of 780 stop signs. The test set is composed of 270 normal stop signs and 20 adversarial examples. We pre-process the data by removing 10% of the border around each sign, and resize every image to $32 \times 32$ pixels.

**Network architecture**. We use the same network architecture settings as our setting for CIFAR-10.

**Results**. Table 8 shows the results of several strong models in Table 1. The results of GAN-based models are absent as [68] observed that GANs did not converge due to the small dataset size. Our HRN again outperforms the best performing baselines significantly, which indicates that our HRN is more robust than others. This is significant in practical applications.

Table 8: Average AUCs in % per method on GTSRB stop sign adversarial attacks

| **Method** | DSVDD | ICS | TQM | HRN |
|------------|-------|------|------|---------|
| AUC | 80.3 | 84.6 | 87.6 | **95.4** |

## 6.2 Using a Pre-trained Feature Extractor

It has been shown that pre-trained feature extractors (or models) can substantially improve the end task results in many image [77] and natural language processing (NLP) applications. For example, pre-trained (language) models (feature extractors) have led to several major breakthroughs in NLP. The most notable such models are BERT [17], XLNet [88], RoBERTa [54], and ALBERT [46].

We experiment with a pre-trained feature extractor using ImageNet. In this case, we use CIFAR-10 as the experimental dataset. ImageNet pre-training is not suitable for the other two datasets as they contain different types of images than ImageNet (they also work very well without pre-training). We pre-train a WRN model [90] using ImageNet after manually removing classes from ImageNet that are similar to those in CIFAR-10. After removal, we are left with 771 classes. The pre-trained feature extractor outputs 640 features.

We experiment pre-training with our HRN system and three top performing baselines. The average result over 10 classes for each system is reported in Table 9. To our great surprise, pre-training helps improve HRN drastically, from 71.32 (Table 1) to 96.6 (Table 9). HRN has a very simple network and only one class of data, which may explain why pre-training is so helpful. ICS also improves greatly, from 69.27 to 86.6. The improvement for TQM is small, from 52.10 to 53.5. OCGAN drops slightly. As reported in [32], pre-training may not always help.

Table 9: Average AUCs in % on CIFAR-10 using pre-training.

| Method | OCGAN | ICS | TQM | HRN |
|--------|-------|------|------|---------|
| AUC | 64.8 | 86.6 | 53.5 | **96.6** |