[Reviews · NeurIPS 2020]

Review 1

Summary and Contributions: The paper proposes a method called HRN for one-class classification / anomaly detection. The loss consists of two parts: a negative log likelihood (NLL) portion and a “holistic” regularizer, which supposedly alleviates some issues with the choice of the NLL loss.

Strengths: The regularizer functions almost like an L2 regularizer that takes into account both the activation and the gradients. Results show some improvements over existing methods.

Weaknesses: The method is simplistic, yet places too much constraints on the type of activation functions that can be used (only ReLU-like activations). Experimental results show incremental improvements on saturated problems with small number of classes. I would be curious to see the results more complex datasets.

Correctness: The proposed method appears correct, but on a higher level I might not use chose the NLL to model the class. The role of the regularizer should be a means to define/control/limit the metric.

Clarity: I’m not a fan of putting the activation function (sigmoid()) in Eq.3, instead of the probability distribution P(C|x). I think page 5 serves little purpose and can be summarized into 1 short paragraph, and the rest of the details can be put into the supplementary material, while the sections in the supplementary material should be within the main text.

Relation to Prior Work: The related work section lists and describes a bunch of papers in the broad field, but has little details to show exactly how these existing methods are related, and how specifically the proposed regularization is related.

Reproducibility: Yes

Additional Feedback: === after rebuttal === After reading the rebuttal and the comments by fellow review, I have gained a better appreciation for this work. I have adjusted my scores to reflect this.


Review 2

Summary and Contributions: This paper proposes a new method in one-class learning (or anomaly detection) where in the training stage, only data from one class is given, but in the test stage, there could be data from other classes or anomalies in addition to the data from the one class given in the training stage. The proposal is simple and has two components: a negative log likelihood loss and a regularization term taking the gradient of the model. The paper gives some examples under two layer MLPs with ReLU activations, and explains that the proposed method aims to have sigmoid output close to 1 while keeping the output of the model as small as possible.

Strengths: - I like the simplicity of the proposed method. - The experiments are encouraging, showing how the proposed method can achieve good performance across a range of datasets. - The paper is organized and easy to follow.

Weaknesses: - The proposed method is based on negative log likelihood, based on empirical risk minimization. This causes to learn extreme outputs since we do not have the anomaly data during training. I was wondering if other loss functions were considered. For example, squared loss may achieve a value of zero unlike NLL, and learning may slow down automatically after a while. I was curious if other loss functions have less issues with Problem-1 (uncontrollable f(x) output). Since the proposed learning objective is simple, it may be nice to explore other options (or explain intuition for choosing log loss). - Are there intuition on why the proposed method is expected to work better (and works better in experiments) than previous one-class methods? I'm guessing the core contribution of the paper is the observation of Problem-2 (feature bias), since Problem-1 (uncontrollable f(x) output) is usually already taken care of in previous one-class classification methods. Is it possible to compare the value of H-regularization term for other methods to see this more clearly? (I think L2 regularization already has this value reported in page 6 which is helpful to see why L2 reg+NLL is not enough for one-class classification.) - According to the ablation study in Table 3, it seems that the normalization step is important for good performance. Without this, NLL+Hreg underperforms baselines in most cases. Is it possible to compare baselines+2N_Inst_Norm and proposed method too? This will make the contributions more clear.

Correctness: yes

Clarity: yes

Relation to Prior Work: Section 2 explains prior work.

Reproducibility: Yes

Additional Feedback: The below are minor questions and minor suggestions: - The chosen architecture seems to be quite small (e.g., 784-100-1), but does this make it disadvantageous for the proposed method compared with others, since others report results from original papers? - A related question is, is this expressive enough to cause the Problem-1 in page 3 to occur? - Is grid search performed for 21 \times 20 = 420 combinations? Although it is fixed for all datasets, I feel this really exploits the negative class information and incorporates it into the training stage. - A minor suggestion is to show how the performance is sensitive to the hyper-parameter of n. - I couldn't see which baselines were using the same results previously reported in original papers and which were reproduced this time. -------------------------------------------------- -------------------------------------------------- After rebuttal period: Thank you for answering all of my questions. Many of my initial concerns are resolved now, and I appreciate the authors for showing additional experimental results to answer my question. It is good to know that the proposed method is useful regardless of the chosen loss function. I have raised my score to take this into account. If accepted, I would like to suggest to the authors to include the discussions and new experiments showing other loss functions such as squared loss in the camera ready version.


Review 3

Summary and Contributions: This paper introduces a novel method HRN to learn a deep one-class classification model. Different from the existing methods that uses classic one-class svm/svdd or adversarially trained one-class model, HRN is trained using a new loss function with the combination of a negative log likelihood and a gradient penalty specifically design for one-class classification problem. Both theoretical and empirical analyses are given to provide key insights into the proposed method HRN.

Strengths: 1). The formulation of HRN is novel and provides a new perspective for devising deep one-class classification model. The novelty and the insights of the approach are clearly presented. 2). The theoretical analysis of HRN well convinces why the method works. 3). The empirical results are complete and convincing. Particularly, extensive empirical results are provided on both image and tabular datasets, and HRN is compared with a number of recent state-of-the-art methods. The experiment settings are good, and HRN generally performs much better than the competing methods. The ablation study also clearly shows the importance of each algorithmic module. I particularly like the experiments in Section 4.5, which show some rarely explored aspects of the deep one-class models, including robustness w.r.t. anomaly contamination and adversarial attacks, and the significance a pre-training may have.

Weaknesses: Some aspects that need to be improved are as follows. 1). It is not precise to state that the methods based on auto-encoders, GANS, self-supervised classification are one-class learning approaches. The objective functions in these methods are very different from the one-class learning objective. I would suggest to rephrase statements similar to this one to avoid misleadings. The recent survey paper "Deep Learning for Anomaly Detection: A Review." arXiv preprint arXiv:2007.02500 (2020). provides some helpful discussions on this matter. 2). in lines 156-159, the authors indicate that the gradient penalty used in HRN is different from that in WGAN and state the difference can result in huge performance gap in their experiments. I believe the authors should provide these results in the supplementary materials and add discussions to help readers fully understand the key difference and the insights of the H-regularization. 3). Traditional state-of-the-art methods like iForest are important baselines and should be included into the empirical comparison. Comparing to them empirically would allow readers to understand what benefits deep anomaly detection may bring. 4). It is interesting that HRN using a MLP architecture can perform better than the competing methods using CNN architectures. It would have more added values if the authors could discuss why this would happen. 5). The performance of HRN on the two tabular datasets is very good. It would be even more convincing if more results on tabular datasets could be added.

Correctness: The method is good, and the claims are well supported by the experiments.

Clarity: The paper is well written. I really enjoy reading this paper.

Relation to Prior Work: Some key references are missing. The authors' statement that most deep anomaly detection methods are for image data is incorrect. There have been a large number of deep anomaly detection methods for non-image data, too. Some recent ones are as follows: "Anomaly detection with robust deep autoencoders." In Proceedings of the 23rd ACM SIGKDD International Conference on Knowledge Discovery and Data Mining, pp. 665-674. 2017. "Learning representations of ultrahigh-dimensional data for random distance-based outlier detection." In Proceedings of the 24th ACM SIGKDD International Conference on Knowledge Discovery & Data Mining, pp. 2041-2050. 2018. "Deep anomaly detection with deviation networks." In Proceedings of the 25th ACM SIGKDD International Conference on Knowledge Discovery & Data Mining, pp. 353-362. 2019. "One-class adversarial nets for fraud detection." In Proceedings of the AAAI Conference on Artificial Intelligence, vol. 33, pp. 1286-1293. 2019. Similar to HRN, these studies handle data with MLP network backbones, and thus, they seem to be more relevant to many other methods in the Related Work that deal with image data.

Reproducibility: Yes

Additional Feedback: Comments made after rebuttal The added results look good. The authors are suggested to insert these new results into the camera-ready version, and to extend the related work as suggested above.

[Author Response · NeurIPS 2020]

**To Reviewer1:** *1. Method simplistic, places too much constraints on activation (only ReLU-like activations).*

We believe the proposed H-regularization is novel and by no means simplistic. It is well suited for one-class learning. ReLU-like activations are widely used, e.g., Transformer, Resnet, etc. It does not affect the application of our method.

*2. I would be curious to see the results more complex datasets.*

In our experiments, we followed baselines and used the same datasets as them. Per your request, we conducted a new experiment using CIFAR100, which has 100 classes. The average accuracy of iteratively taking each class as the one class to do one-class learning is 68.61 which is much better than top baselines' (52.26 for OCGAN, 60.79 for ICS).

*3. I'm not a fan of putting the activation function (sigmoid()) in Eq.3, instead of the probability distribution P(C|x).*

Since we have only one class in the output, we believe using sigmoid() to represent the score of x belonging to the one class is reasonable. It is hard to estimate the probability distribution P(C|x) without other (negative) data.

*4. page 5 can be summarized into 1 paragraph ...* Thanks for the suggestion. We will follow it in revising our paper.

**To Reviewer2:** *1. ... if other loss functions (e.g., squared loss) were considered, which may not have Problem-I ...*

We experimented with squared loss and found it also faces the saturation problem as the output targets are always 1 during training. It gets quite poor results, 71.35 (on MNIST). A small learning rate 0.01 did not help. If we add our H-regularization and normalization method, the result gets to 97.38, which is quite close to our result (97.59) using the NLL loss. This indicates that our H-regularization and normalization method are not limited to NLL.

*2. ... why the proposed method works better ... the core contribution ... is the observation of Problem-2 (feature bias).*

The intuition is that our method tries to leverage features holistically to ensure the system is not biased towards certain features. This decreases the probability of abnormal (negative) data passing the system to achieve better results. It is hard to compare the value of the H-regularization term in baselines as they have completely different loss functions which make the H-regularization values incomparable.

*3. Is it possible to compare baselines+2N_Inst_Norm and proposed method too? It will make the contributions clearer.*

We experimented with replacing the normalization method of the top baselines ICS, TQM and OCGAN with ours but got quite poor results. This is because each baseline already has its most suitable normalization method for its approach.

*4. The chosen architecture is quite small (e.g., 784-100-1), but does this make it disadvantageous ... ?- A related question is, is this expressive enough to cause the Problem-1 in page 3 to occur?*

It can be disadvantageous to us. It may also mean that our method still has room for improvement. We did try a more complex CNN architecture, but the improvement is small. We will investigate more in our future work. Simple architecture also causes Problem-I as we detected saturated outputs too (hope we understood your question correctly).

*5. Is grid search performed for $21 \times 20 = 420$ combinations? Although it is fixed for all datasets, ...*

It is $21 + 20 = 41$ (we try one, fix it and then next). We should note that the hyper-parameter search is only done on MNIST. The resulting parameters are used for all the other datasets, i.e., no parameter tuning needed for each dataset.

*6. A minor suggestion is to show how the performance is sensitive to n.* When $10 \leq n \leq 16$, all the results are good.

*7. which baselines were using the same results previously reported in original papers.*

The results of OCGAN on the FMNIST data and TQM on the CIFAR10 data are obtained by running the author released code and the rest are copied from published papers. We will detail this in Appendix.

**To Reviewer3:** Thanks for your constructive suggestions. We will improve accordingly and cite the papers you listed.

*1. iForest is an important baseline and should be included into the empirical comparison.*

We run sklearn's API. iForest gets 94.74% and 94.44% (F1 score) on KDDCUP and Thyroid respectively. And the AUCs of iForest on MNIST/FMNIST/CIFAR10 are 84.43/90.51/59.70. Our method ourperforms all of them.

*2. It is interesting that HRN using a MLP can perform better than ... the authors could discuss why this would happen.*

We believe one of the key issues in one-class learning is how to avoid biasing some features as there is no negative data. We did not see that existing approaches explicitly deal with this problem. In our case, we identify and explicitly address the model bias problem using H-regularization, which we believe is the main reason.

*3. HRN on the 2 tabular datasets is very good. ... more convincing to add more*: We will add more datasets. We ran another dataset **Arrhythmia** and obtained (F1): 45.8 (OCSVM), 49.8 (DAGMM), 53.0 (TQM), and 84.5 (**HRN**).

[Meta-Review · NeurIPS 2020]

This paper proposes a novel deep one-class classification method where a regularization technique is specially designed for one-class classification problem. It also provides insights on the bottlenecks of previous methods for this problem; one insight is quite novel and has not been considered yet (representation learning from one-class data is biased to the given training data), since previous methods mainly focused on the other (deep network outputs become over-confident given one-class data). I am feeling the paper may further inspire more cleverly designed methods for this problem! While in the beginning the reviewers had some concerns (mainly the clarity and the generality that is related to the significance), the authors did a particularly good job in their rebuttal (showing that the proposal is not limited to a single surrogate loss function). Thus in the end, all of us have agreed to accept this paper for publication! Please carefully address the concerns from all 3 reviewers in the next version.